# Assessing the Potential of HPV16 E6 Seroprevalence as a Biomarker for Anal Dysplasia and Cancer Screening—A Systematic Review and Meta-Analysis

**DOI:** 10.3390/ijms25063437

**Published:** 2024-03-19

**Authors:** Sara Tous, Mariona Guillamet, Tim Waterboer, Laia Alemany, Sonia Paytubi

**Affiliations:** 1Cancer Epidemiology Research Programme, Catalan Institute of Oncology-IDIBELL, Av Granvia de l’Hospitalet 199-203, 08908 L’Hospitalet de Llobregat, Spain; stous@iconcologia.net (S.T.); mguillamet98@gmail.com (M.G.); lalemany@iconcologia.net (L.A.); 2Consortium for Biomedical Research in Epidemiology and Public Health—CIBERESP, Carlos III Institute of Health, Av. De Monforte de Lemos 5, 28029 Madrid, Spain; 3Division of Infections and Cancer Epidemiology, German Cancer Research Center (DKFZ), 69120 Heidelberg, Germany; t.waterboer@dkfz-heidelberg.de

**Keywords:** human papillomavirus, anal cancer, anal dysplasia, HPV16 E6, biomarkers, serology

## Abstract

Elevated rates of human papillomavirus (HPV)-related anal high-grade squamous intraepithelial lesions (HSIL) and anal cancer (AC) in populations like men who have sex with men (MSM) living with HIV underscore the need for effective screening. While high-resolution anoscopy-guided biopsy is the gold standard, limited provider availability poses a challenge. This has spurred interest in identifying biomarkers for improved AC prevention. Antibodies against HPV16 oncoprotein E6, known as markers for cervical and oropharyngeal cancers, are the focus of the current study. The systematic review and meta-analysis included six studies meeting inclusion criteria, assessing HPV16 E6 seroprevalence in individuals with anal HSIL or AC. A two-step meta-analysis estimated pooled odds ratios and 95% confidence intervals (CI) for HPV16 E6 seroprevalence and HSIL or AC. Pooled prevalence, sensitivity, specificity, and diagnostic odds ratios were also calculated. This meta-analysis revealed a 3.6-fold increased risk of HSIL for HPV16 E6 seropositive individuals, escalating to a 26.1-fold risk increase for AC. Pooled specificity and sensitivity indicated a high specificity (0.99; 95%CI: 0.99, 0.99) but lower sensitivity (0.19; 95%CI: 0.10, 0.34) for HPV16 E6 serostatus as an AC biomarker. In conclusion, while HPV16 E6 seroprevalence demonstrates specificity as a potential biomarker for HPV-related AC, its utility as a standalone screening tool may be limited. Instead, it could serve effectively as a confirmation test, particularly in high-risk populations, alongside other diagnostic methods. Further research is imperative to explore HPV16 E6 seroconversion dynamics and alternative screening algorithms.

## 1. Introduction

Anal cancer (AC) is a rare disease in the general population. However, it is the second most frequent non-AIDS-defining cancer and its incidence in high-risk populations is currently similar or even higher (85 per 100,000 person-years) [1] to that of cervical cancer in the female general population of high-income countries before the introduction of cervical cancer screening programs [1,2,3]. Human papillomavirus (HPV) is the causal agent of most cases of AC, specifically high-risk oncogenic HPV genotypes (HR-HPV) [4,5]. 

AC has a well-described precursor lesion, a high-grade squamous intraepithelial lesion (HSIL). Both HSIL and AC can be detected early with screening programs. Among the general population, the prevalence of AC is low and does not justify routine screening. However, the prevalence of anal cancer is significantly higher in specific populations and the recommendations in guidelines focusing on these high-risk individuals need particular attention. The recently released recommendations from the International Anal Neoplasia Society (IANS) represent a significant milestone, as they are the first guidelines that offer evidence-based consensus guidance on AC screening. These long-awaited guidelines address a critical need within the scientific and medical community, providing clarity and direction in an area where standardized recommendations were previously lacking [6]. Multiple diagnostic tools with various sensitivities and specificities have been evaluated for screening patients at risk for anal cancer, including anal cytology, high-resolution anoscopy (HRA), high-risk HPV genotyping, HPV and host methylation markers and HPVE6/E7 mRNA [7,8,9,10,11,12]. The gold standard for diagnosing anal lesions is HRA followed by an anal biopsy. However, these procedures come with some limitations. Their invasive nature, the need for specialized equipment and expertise, the potential discomfort for patients, and the possibility of bleeding or other complications, can increase treatment morbidity. Additionally, HRA has a high cost and may not be readily available, particularly in low-resource settings [13,14]. Additionally, while anal cytology has a high specificity for detecting HSIL, its sensitivity is too low to be used as a standalone screening test. Furthermore, while anal HPV testing is more sensitive compared to cytology, its low specificity, particularly in high-risk populations such as men who have sex with men (MSM) living with HIV, results in a benefit only for those who test negative [15]. Considering the growing burden of AC in high-risk populations, there is a need for research on new, specific, non-invasive HPV-related biomarkers to improve screening programs. 

Most HSILs will never progress to AC. In a meta-analysis study, the risk of progression from anal HSIL to cancer was estimated to be 265 per 100,000 person-years among MSM that are living with HIV [16], which is a progression rate lower than for cervical pre-neoplastic lesions. Yet, a recent Danish report estimated a 5-year risk of AC following a diagnosis of HSIL of 14.1% among people living with HIV [17]. Therefore, the identification of biomarkers that could predict the progression of low-grade squamous intraepithelial lesion (LSIL) or HSIL to AC would be very useful. This could be of pivotal importance to detect those lesions at the highest risk of progressing to AC and thus, only those patients would receive treatment. Consequently, this will offer the opportunity to design more cost-effective AC screening algorithms. 

During HPV infection, early (E1, E2, E4, E6, and E7) and late (L1 and L2) HPV genes are transcribed throughout the virus life cycle. The early HPV proteins reflect productive infection of epithelial cells, where the action of the E6 and E7 oncoproteins and their partner host proteins p53 and pRb, reactivates cell division, inhibits apoptosis, and abrogates epithelial differentiation [18,19,20,21]. Thus, these proteins are known to be involved in the suppression of tumor growth and have been linked to the development of malignant epithelial transformation. For that reason, alternative biomarkers are being explored, such as antibodies to the oncoproteins HPV E6 and E7. 

The aim of this meta-analysis was to evaluate the performance of HPV16 E6 seroprevalence for the detection of HSIL and AC, which is of special interest in high-risk populations. Herein, HPV16 E6 seropositivity has been found to be a specific marker with potential clinical significance for a triage strategy.

## 2. Materials and Methods

### 2.1. Search Strategy and Literature Selection

A systematic literature review without language restrictions for published studies of HPV16 E6 seropositivity in anal HSIL and AC was performed. We conducted the systematic review and meta-analysis following the Preferred Reported Items for Systematic Reviews and Meta-Analyses (PRISMA) guidelines [22]. A protocol was prepared for this study, although it was not registered to any specific registry. Search criteria involved publication up to 1 January 2024. The search was made through PubMed using the following combined search strategy: (“hpv” OR papillom*) AND ((serop*) OR antibod*) AND (“anal cancer” OR “anal HSIL”) without language restrictions. The screening process was performed by two investigators (ST and SP), independently. Any disagreements pertaining to the inclusion or exclusion of studies were addressed through discussion. Articles were selected by first screening the titles and abstracts, and then screening the entire publication. Reference lists of retrieved articles were reviewed to identify additional relevant studies. Case–control and cohort studies with anal HSIL+ as the endpoint were included. Anal HSIL+ comprises two high grades of anal intraepithelial neoplasias (AIN2 and AIN3), ASC-H (atypical squamous cells that do not exclude HSIL), and any type of anal cancer. Studies having quantitative estimates for the variables of interest and number or percentage of patients with HPV16 E6 seropositivity results for both cases and controls were included. Moreover, only serum samples from the different studies that were tested for HPV16 E6 antibodies by using multiplex serology based on glutathione S-transferase fusion proteins in combination with fluorescent bead technology [23] were included. Studies that evaluated HPV16 seroprevalence for other antigens than the HPV16 E6 oncoprotein were excluded. Excluding other antigens ensures a narrow focus, enabling a precise investigation into HPV16 E6 seropositivity, linked to the risk of HPV-driven neoplastic processes in cervical, oropharyngeal cancers, and anal cancer, in contrast to antibodies targeting other HPV proteins [24] (Appendix A). Data from abstracts, unpublished studies, reviews, meta-analyses, case reports, and commentaries were excluded.

### 2.2. Data Extraction

Articles that matched the inclusion criteria were extensively analyzed by two investigators to extract the HPV16 E6 seroprevalence for cases and controls. Additionally, authors were contacted three times to provide information if not included in the of original publication. Study details were tabulated and, for every study, name of the first author and year of publication, country of the cohort and study location, age range and gender, HIV status, sample size, lesion endpoint, serologic assay, and time of the serum collection as well as estimates of the seroprevalence and odds ratio (OR), whether adjusted or not, were recorded. 

Multiplex serology shows median fluorescence intensity values that are dichotomized as antibody positive or negative, using established seropositivity cutoffs. The cutoffs used in all studies were 484 MFI, based on a previous study [25], with the exception of Combes 2017 [26], which used a cutoff of 1000 MFI. 

### 2.3. Statistical Analysis

A two-step meta-analysis was used to estimate pooled OR and their corresponding 95% CI between HPV16 E6 seroprevalence and HSIL+ anal lesions. Due to the presence of variability among the populations sampled, the global effect estimation was predicted by using a random effects model with Stata *meta* command. Firstly, a random effects model with treatment arm continuity correction for Bertisch study [27] was performed [28]. This approach was undertaken to derive crude OR estimates due to the absence of reported cases positive for HPV16 E6 [27] in the original analysis, resulting in an infinite OR value (Appendix A). Then, a second random effects meta-analysis was performed to calculate the pooled OR estimates using the restricted maximum likelihood (REML) estimator [29]. Forest plots were used to present the results. Heterogeneity and inconsistency between studies were assessed using the I^2^ statistic and *p*-value for heterogeneity and the quantification of between-study variance using the Τ^2^ (*tau*^2^). Evaluation of the influence of each study on the global effect estimation was conducted by recalculating the overall effect estimate after omitting each study using the *metaninf* function. Begg’s and Egger’s tests were performed to identify the risk of publication bias employing *meta bias* function and graphically visualized using funnel plots and Trim and fill method. Pooled prevalence, sensitivity (Se), specificity (Sp), positive and negative likelihood ratios, and diagnostic odds ratios (DOR) were also calculated. All analyses were stratified by endpoint (AC or HSIL lesion). Two sensitivity analyses were performed, one excluding Bertisch [27] to better measure the magnitude of the effect and another excluding Combes (2017) [26] due to the different cutoff in MFI used (Appendix A). Methodological quality was assessed using the Appraisal Tool for Cross-Sectional Studies (AXIS). All the analyses were run with Stata SE version 16 (StataCorp LLC, TX, USA).

## 3. Results and Discussion

### 3.1. Literature Selection and Study Characteristics

The initial search from the electronic database identified 141 studies (Figure 1). Of those, 124 records screened by title and abstract, did not meet the eligibility criteria. Seventeen full-text studies were then retrieved. However, after further scrutiny, 11 of these studies were subsequently excluded due to the following specific reasons: eight studies did not perform HPV E6 serology, two studies lacked an anal HSIL+ endpoint, and one study did not provide HPV16 E6 seroprevalence data for cases and controls. A total of six articles [26,27,30,31,32,33] that evaluated the association between HSIL+ lesions and HPV16 E6 seroprevalence, in a total of 13,144 participants, were eligible to be included in the analysis. None of the studies reported the HPV vaccination status of the individuals included in the study of HPV16 E6 seroprevalence. The six studies were published between 2013 and 2020. A description of the demographic and clinical characteristics of the studies is presented in Table 1. 

### 3.2. Meta-Analysis of HPV16 E6 Seropositivity in HSIL+ Patients and Publication Bias

The global effect estimate, determined through a random effects meta-analysis, showed a 14.56 times higher probability of HPV16 E6 seropositivity in patients with HSIL+ compared to controls (95%CI: 5.25, 40.35) (Appendix A). These findings underscore the significant association between HPV16 E6 seropositivity and the presence of high-grade anal lesions. However, differences in the estimates emerged when distinguishing between HSIL and AC as distinct endpoints (Figure 2). The designated endpoint differed across the studies. Specifically, AC served as the endpoint for Bertisch [27], Kreimer [30], Combes (2017) [26], and Combes (2020) [32]. On the other hand, Poynten [33] adopted composite HSIL (identified through cytology and histology) as the designated endpoint. Karita [31], however, considered both HSIL and AC as endpoints. Further stratified analysis demonstrated a 26.09-fold increase (95%CI: 8.70, 78.28) in the probability of developing AC in individuals testing positive for HPV16 E6, as compared to controls (Figure 2A). The most influential study was Karita [31], carrying the largest weight percentage among studies at approximately 28%, followed by Combes (2020) [32] and Kreimer [30], contributing 26% and 21%, respectively. The statistical significance of the pooled estimate, indicated by a *z*-test with a *p*-value less than 0.001, emphasized its significant deviation from 0. The observed heterogeneity (I^2^ = 66%) denoted a moderate level of variability within the effect size estimates, primarily due to the between-study differences rather than mere sampling variation. Τ^2^ reached 0.93, while the homogeneity test exhibited a significant *p*-value (*p* = 0.02), thereby providing definitive statistical evidence of the between-study heterogeneity. Furthermore, upon evaluating the influence of each study, it was found that Combes’ (2020) study exhibited an excessive influence (Appendix A). The exclusion of this study resulted in a recalculated pooled estimate of 40.78 (95%CI 20.95, 79.39). The observed variations in OR estimates among studies could potentially be attributed to population-specific factors, such as gender distribution, case-to-control ratio, and cohort size. As previously discussed, the study conducted by Combes (2020) [32], which had the lowest case-to-control ratio (0.8% cases), could have significantly impacted the overall estimate. Moreover, the authors of this particular study postulated that the propensity for seroconversion in the Swiss HIV cohort is more pronounced among women compared to men. This premise implies that the incorporation of a diminished proportion of women within a study would lead to lower OR estimates, while a higher proportion of women would yield higher estimates. Consequently, the inclusion of Combes (2020) [32], which featured a large cohort, a low case-to-control ratio, and a relatively low proportion of women, contributed to the reduction in the overall estimate. 

Publication bias was assessed for AC using funnel plots (Appendix A). The plots revealed a lack of studies in the middle left of the graph, and if present, a paucity of studies in the lower-left section, indicating a dearth of smaller studies reporting small effect sizes. Despite these observations, the assessment of asymmetry using Egger’s regression-based test did not yield any discernible evidence of such asymmetry (*z* = 0.74; *p*-value = 0.459). However, it is important to acknowledge that the power of this method was limited given the small number of studies included in the analysis. Upon application of the trim and fill method, two additional studies were imputed, resulting in an adjustment of the overall effect size estimate to 17.01 (95%CI 6.36, 45.49) (Appendix A). 

The analysis revealed lower OR in the two studies that included patients with HSIL instead of AC as the endpoint (pooled OR = 3.60, 95%CI: 1.44, 9.03) (Figure 2B). The likelihood of HPV16 E6 seropositivity in individuals with anal HSIL was of borderline significance, with a lower OR observed in the study conducted by Poynten [33] (OR = 2.97, 95%CI 0.92, 9.59) compared to the other studies. Moreover, this risk is consistent with the findings found by Karita [31] (OR = 4.9, 95%CI 1.12, 21.51) when only HPV16 E6 seroprevalence in HSIL patients is considered. Furthermore, statistical evidence emerged to conclude that the pooled estimate for HSIL lesions surpassed the null hypothesis (*z* = 2.73, *p*-value = 0.01). Notably, no heterogeneity was observed in the HSIL analysis, and the evaluation of publication bias reported a missing study that, when accounted for, reduced the pooled estimate to 2.97 (95%CI 1.36, 6.48) (Appendix A).

The outcomes derived from the current study underscore heightened odds of AC in patients exhibiting HPV16 E6 seropositivity in contrast to those with HSIL. As demonstrated in some studies, this suggests a potential occurrence of HPV16 E6 seroconversion in closer temporal proximity to the diagnosis of AC [30]. Nevertheless, further extensive research is needed to determine the precise timing of HPV16 E6 seroconversion ahead of AC diagnosis. Additionally, investigation into potential gender-based disparities in HPV16 E6 seroprevalence, especially in high-risk populations, necessitates longitudinal studies for a more comprehensive understanding. 

### 3.3. Variability in HPV16 E6 Seroprevalence Studies among Patients with HSIL+

The seroprevalence of HPV16 E6 among patients with incident AC was evaluated. According to Karita [31], the proportion of individuals with incident AC who presented HPV16 E6 seropositivity was 29%, which is consistent with the sensitivities shown in Kreimer (29%) [30], Combes (2017) (33%) [26] and to a lesser extent, Bertish (22%) [32]. However, a notably lower seropositivity rate of 5% was found in Combes (2020) [32]. Furthermore, as previously mentioned, Combes (2020) [32] reported a higher likelihood of seroconversion among women compared to MSM, both living with HIV. Although this difference was only marginally significant, it suggests that the sensitivity and negative predictive value (NPV) of HPV16 E6 serology within the context of HIV-positive MSM individuals—a demographic at a higher risk for AC—may be lower. This could also explain why seropositivity in individuals with AC in the Swiss HIV Cohort Study (SHCS) from Bertisch and Combes (2020) [27,32], which primarily includes HIV-positive men, is lower in comparison to that in Kreimer [30] from the European Prospective Investigation into Cancer and Nutrition (EPIC) study that primarily includes HIV-negative women. However, in Combes (2017) [26], which only included HIV-positive MSM from the SHCS, seropositivity was higher, nevertheless, the case-to-control ratio was very low. Despite these observations, no evidence of a relationship between HIV-related immunosuppression (CD4 cell counts) and HPV16 E6 seropositivity in individuals with AC has been found to date. 

On the other hand, Poynten [33] focused on HSIL cases from HIV-positive and HIV-negative gay and bisexual men (GBM) in the context of the Study of Prevention of Anal Cancer (SPANC). This study revealed a low prevalence of HPV16 E6 seropositivity (3.8%), which may be attributed to several factors not mutually exclusive, including the restriction to male participants, the HSIL endpoint, and potential disparities in sexual behavior. Similarly, in the study conducted by Karita [31], the prevalence of HPV16 E6 seropositivity stood at 4.5%. This study involved a cohort of individuals with histologically confirmed anal HSIL, as diagnosed through the Cancer Surveillance System in Washington. Notably, women constituted the majority among both the cases and the control groups in this study. 

Moreover, the timing of serum sample collection in relation to the diagnosis of HSIL or AC varied across studies, with some collection of samples close to diagnosis and others after diagnosis (Table 1). These temporal deviations in sample collection might significantly contribute to the observed differences in HPV16 E6 seroprevalence outcomes across studies. Remarkably, some studies have shown that HPV16 E6 seroprevalence remains low more than 2–5 years preceding AC diagnosis [30,32]. Furthermore, approximately 20% of HPV16 E6-seropositive individuals diagnosed with AC show at least one subsequent seronegative sample, including, for the majority, their last sample prior to their anal cancer diagnosis [32]. 

In summary, the variations observed between studies could be attributed to a variety of factors, including the specific endpoints and populations under investigation such as people living with HIV, the utilization of antiretroviral therapy, gender composition, sexual behavior, as well as the potential confounding effect of concurrent HPV16 infections at other anatomic sites. 

It is important to note that the specificity of HPV16 E6 as a marker for HPV-related AC was confirmed in all studies, with low seroprevalence in the control group ranging from 0% to 1.31%. Contrarily, a low sensitivity for HPV16 E6 seroprevalence was observed across the studies, particularly in the two studies of HSIL [31,33] (Table 2). In general, the negative predictive value (NPV) was found to be higher than the positive predictive value (PPV) and was deemed more informative in this context, given that the high specificity of the test provided improved assurance of a low risk of HPV-driven AC for those who tested HPV16 E6 seronegative. 

### 3.4. Diagnostic Performance of HPV16 E6 Seroprevalence

The analysis of studies that included HSIL+ as the endpoint provided a pooled sensitivity and specificity of HPV16 E6 seroprevalence of 0.12 (95%CI 0.06, 0.24) and 0.99 (95%CI 0.99, 0.99), respectively (Appendix A). The pooled diagnostic odds ratio (DOR) was 13 (95%CI: 5, 32), and the positive and negative diagnostic likelihood ratios (DLR) were 11.6 (95%CI 5.3, 25.4) and 0.89 (95%CI 0.80, 0.98), respectively, indicating that the HPV16 E6 serological test can provide confirmation but not exclusion of HSIL+. There was considerable heterogeneity among studies, as demonstrated by the statistically significant chi-squared test result (*p* < 0.01) and an I^2^ value of 89% (95%CI 78, 100). On the other hand, the pooled sensitivity and specificity of HPV16 E6 serology when only AC cases were considered were 0.20 (95%CI 0.10, 0.34) and 0.99 (95% CI 0.99, 0.99), respectively (Figure 3). The pooled DOR for AC was 23 (95%CI 10, 53), the positive DLR for AC was 19 (95%CI 9.6, 37.5) and the negative DLR was 0.81 (95%CI 0.70, 0.94). These results suggest that for AC a positive serologic test may be used for confirmation only. The heterogeneity among studies was significant, as evidenced by the statistically significant chi-squared test result (*p* = 0.007) and an I^2^ value of 77% (95%CI 49, 100).

One aspect to consider when conducting a meta-analysis is the cutoff value used for the test. Variability in such cutoffs can impact the sensitivity and specificity of the test in each cohort. The cutoff values employed in all studies were set at 481 MFI, except for the study conducted by Combes in 2017 [26], which applied a higher cutoff threshold of 1000 MFI. However, given the small sample size of this study and the limited range of MFI values typically found between 484 and 1000, a sensitivity analysis was performed, resulting in similar outcomes regardless of the inclusion or exclusion of this study (Appendix A). 

### 3.5. Limitations and Future Potentials of HPV16 E6 Seropositivity as a Biomarker for Anal Cancer Risk Stratification and Screening

Non-invasive screening tools are needed for early detection and prognosis of HSIL+, particularly among populations at higher risk of AC. The pooled data of six studies, including 13,144 men and women with a diagnosis of HSIL+ across different geographical regions, has allowed us to systematically estimate the prevalence of HPV16 E6 seropositivity. More specifically, when the meta-analysis was tailored to studies focused on AC, it unveiled a positive association between HPV16 E6 seropositivity and AC occurrence. It is noteworthy that an in-depth examination has revealed HPV16 E6 seropositivity to exhibit a commendable high level of specificity as a biomarker. However, its sensitivity has been observed to be comparatively low, particularly in instances where studies involving HSIL cases are encompassed. Combes and collaborators [32] suggest that repeating serological testing may potentially offset low sensitivity and fluctuating kinetics, thereby enhancing the stratification of AC risk. It is our belief that the use of this biomarker, coupled with a sensitive HPV test, has the capacity to enhance specificity in populations with elevated HPV prevalence such as MSM living with HIV. This improvement could lead to a more efficient triage strategy, subsequently reducing the number of invasive HRA procedures required for the detection of AC. In this context, further research would be needed to confirm the effectiveness of this approach and to determine the optimal combination of these two tests in high-risk populations. 

It is important to note that there are certain limitations of the current study that should be taken into consideration when interpreting the outcomes. These limitations include, though are not limited to, the relatively restricted number of studies included in the meta-analysis, which may impact the statistical power of the analysis and potentially render it susceptible to publication bias. Additionally, as mentioned above, the studies included in the analysis used different endpoints and populations, thereby potentially engendering heterogeneity within the results and potentially constraining the generalizability of the findings. An additional noteworthy observation is the relatively diminished sensitivity of the HPV16 E6 seropositivity test, particularly when the analysis incorporates HSIL cases. To mitigate the potential impact of varying endpoints on the results, a separate analysis was undertaken to include only those studies that utilized AC as the primary endpoint. Additionally, the HPV vaccination status of the individuals involved in the respective studies on HPV16 E6 seroprevalence was not mentioned in any of the articles. Potential co-variations of HPV16 E6 seroprevalence with vaccination status might have had an impact on each individual study and on this meta-analysis.

In light of the limitations of the study, it is important to conduct further research on the relationship between HPV16 E6 serostatus and AC, taking into consideration specific population characteristics such as gender, sexual behavior, and HIV status. This will provide a more comprehensive understanding of the utility of this biomarker in different populations and help inform the development of more effective screening and diagnostic strategies. Further research is also needed to understand the kinetics of HPV16 E6 seroconversion in order to understand the timing of HPV16 E6 seroconversion and to improve the sensitivity of the test. Also, future work in larger populations will need to explore the potential of using HPV16 E6 serostatus as a triage tool for high-risk populations to decrease the number of invasive HRA procedures needed for AC detection. This will help clarify the relationship between HIV-related immunosuppression and HPV16 E6 seropositivity in individuals with anal cancer. It would also be beneficial for the research community to evaluate the cost-effectiveness of utilizing HPV16 E6 serostatus as a screening tool for AC, particularly in low-resource settings as a means of informing future implementation strategies. 

Last, but not least, HPV vaccination has great promise to reduce anal HSIL or AC burden. Indeed, a recent systematic review and meta-analysis [34] has underscored the substantial vaccine efficacy/effectiveness against incident/prevalent HPV infection and associated disease. Nonetheless, the full extent of its impact may not be fully realized for several decades, particularly given the recent initiation of vaccination among men. Thus, the identification of a specific non-invasive biomarker to discern individuals at the highest risk could significantly enhance algorithms for the secondary prevention of anal cancer within high-risk populations.

## 4. Conclusions

Our systematic review and meta-analysis indicates that HPV16 E6 serostatus is a specific biomarker for AC. However, based on the pooled sensitivity reported in this manuscript, it seems that HPV16 E6 seropositivity may not be an ideal standalone test for detecting anal cancer. Nevertheless, it could potentially be used as a confirmation test in combination with other diagnostic methods, or as a triage test to identify individuals at high risk of AC who may warrant further evaluation. It would be important to consider the performance of this test in the specific population of interest, as well as the availability and feasibility of other diagnostic options.

## Figures and Tables

**Figure 1 ijms-25-03437-f001:**
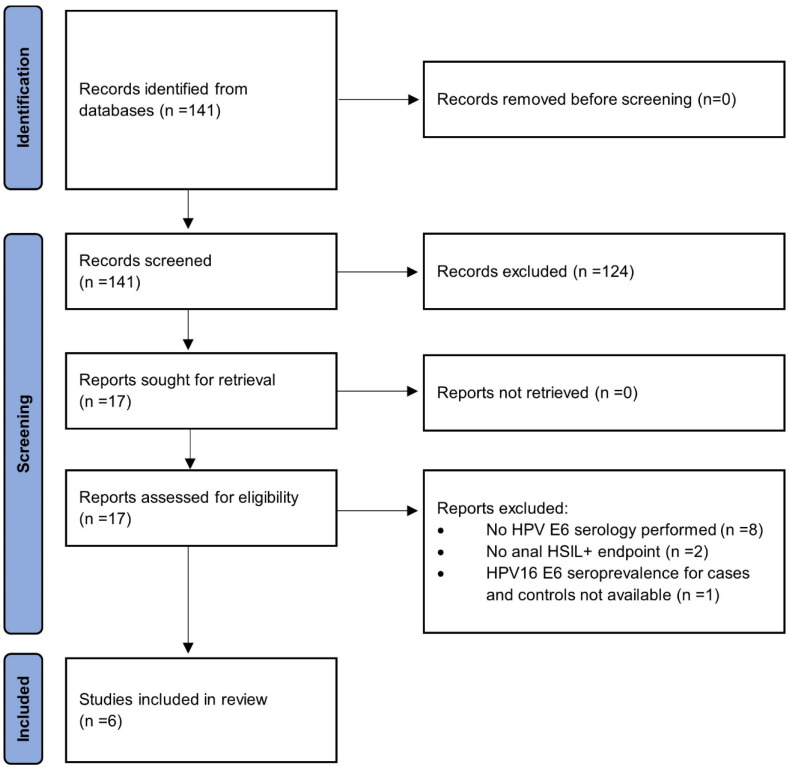
PRISMA 2020 flow diagram representing the systematic review process performed. HSIL+: high-grade squamous intraepithelial lesion or anal cancer. HPV: human papillomavirus.

**Figure 2 ijms-25-03437-f002:**
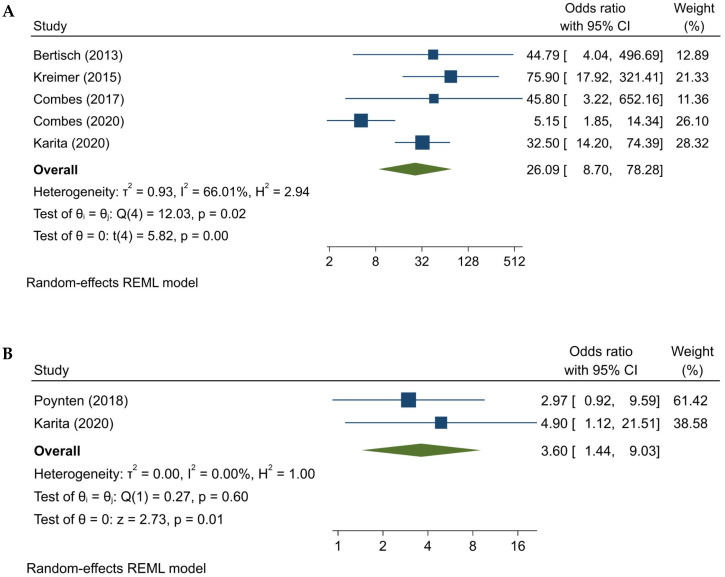
Forest plots of random effects model analyses of the studies included in the meta-analysis assessing the relationship between HPV16 E6 seroprevalence and the risk of AC (**A**) and HSIL (**B**). The odds ratios from individual studies, together with the summary measure (represented by the centerline of the diamond) and their corresponding 95% CI, are depicted. Weights are assigned to individual studies according to their contribution to the pooled estimate, as reflected in the last column and size of the box. The overall degree of heterogeneity in each meta-analysis is indicated by the I^2^. References: Bertisch (2013) [27], Kreimer (2015) [30], Combes (2017) [26], Combes (2020) [32], Poynten (2018) [33], Karita (2020) [31].

**Figure 3 ijms-25-03437-f003:**
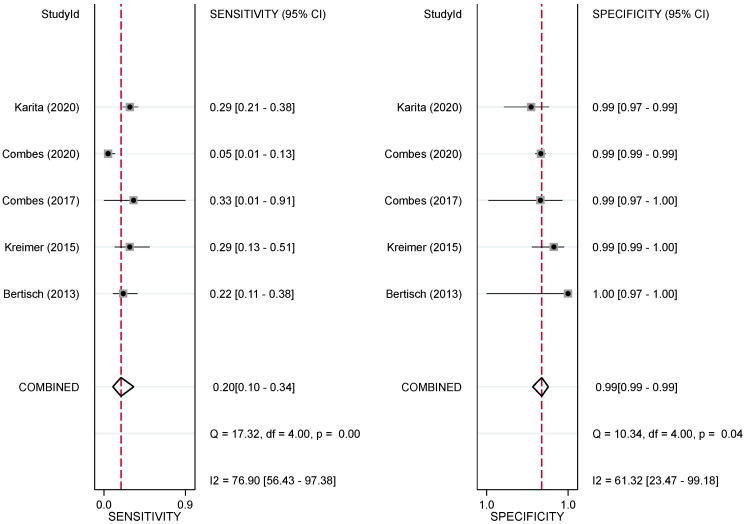
Forest plot indicating the pooled sensitivity and specificity for HPV16 E6 serology for the detection of AC. Each individual study’s point estimate, along with its corresponding 95% CI, is represented by squares and horizontal lines, respectively. Diamonds indicate combined sensitivity and specificity with the red line indicating the combined point estimate. References: Karita (2020) [31], Combes (2020) [32], Combes (2017) [26], Kreimer (2015) [30], Bertisch (2013) [27].

**Table 1 ijms-25-03437-t001:** Description of the characteristics of the studies.

Authors (Year)	Cohort	Age Range (y.o.)	Location	Gender	HIV Status	Sample Size ^a^ (Cases/Controls)	Endpoint	Serologic Assay	Time of Serum Collection	HPV16 E6 Seroprevalence (Cases/Controls)	Effect Measured	OR (95% CI)
Bertisch (2013) [27]	Swiss HIV Cohort Study	≥25	Switzerland	Men (predominant) and women	HIV- positive	155 (41/114)	AC	Multiplex serology assay	At AC diagnosis	21.95/0	Adjusted OR (95%) ^b^	Inf. ^e^
Kreimer (2015) [30]	EPIC	≤40–>70	Europe	Men and women (predominant)	n.s.	743 (24/719)	AC	Multiplex serology assay	Before AC diagnosis	29.16/ 0.28	Adjusted OR (95%) ^c^	75.90 (17.92, 321.41)
Combes (2017) [26]	Swiss HIV Cohort Study	19.6–73.40 (mean age range)	Switzerland	Men (MSM)	HIV-positive	281 (3/278)	AC	Multiplex serology assay	Before AC diagnosis	33.33/ 1.08	Unadjusted OR (95%)	45.83 (3.22, 652.16)
Combes (2020) [32]	Swiss HIV Cohort Study	>25	Switzerland	MSM (predominant), non-MSM and women	HIV-positive	10,384 (76/10,308)	AC	Multiplex serology assay	Before AC diagnosis	5.26/ 1.07	Unadjusted OR (95%)	5.15 (1.85, 14.34)
Karita (2020) [31]	Cancer Surveillance System	18–78	Washington (USA)	Men and women (predominant)	n.s.	946 (116/830)	AC	Multiplex serology assay	After AC diagnosis	29.31/ 0.73	Adjusted OR (95%) ^d^	32.50 (14.20, 74.39)
Poynten (2018) [33]	SPANC	35–79	Sydney (Australia)	Men (GBM)	HIV-positive and HIV-negative	568 (263/305)	HSIL	Multiplex serology assay	n.s.	3.80/ 1.31	Unadjusted OR (95%)	2.97 (0.92, 9.59)
Karita (2020) [31]	Cancer Surveillance System	18–78	Washington (USA)	Men (predominant) and women	n.s.	897 (67/830)	HSIL	Multiplex serology assay	After HSIL diagnosis	4.48/ 0.73	Adjusted OR (95%) ^d^	4.90 (1.12, 21.51)

^a^ Sample size of the patients included in the HPV serology analysis. ^b^ Adjusted OR for the following matching variables: sex, HIV transmission category, age at anal cancer (AC) (years), calendar period at anal cancer diagnosis, duration of the follow-up prior to AC (years). ^c^ Adjusted OR for the following matching variables: country, sex, and age. ^d^ Adjusted OR for age, gender, smoking status, and number of sex partners. ^e^ Infinite OR due to zero controls among nine HPV16 E6 seropositives. n.s.: not specified. MSM: men who have sex with men. GBM: gay and bisexual men. y.o.: years old. OR: Odds ratio. CI: Confidence interval.

**Table 2 ijms-25-03437-t002:** Sensitivity, specificity, predictive positive and negative values of the studies.

Authors (Year)	Endpoint	Raw Sens	Raw Sp	Raw PPV	Raw NPV	Calculated Sens	Calculated Sp	Calculated PPV	Calculated NPV
Bertisch (2013) [27]	AC	0.22	1.00	1.00	0.78	0.22	1.00	1.00	0.78
Kreimer (2015) [30]	AC	0.29	0.99	0.64	0.98	0.29	0.99	0.34	0.98
Combes (2017) [26]	AC	0.33	0.99	0.25	0.99	0.33	0.99	0.25	0.99
Combes (2020) [32]	AC	0.05	0.99	0.03	0.99	0.05	0.99	0.035	0.99
Karita (2020) [31]	AC	0.20	0.99	0.75	0.85	0.29	0.994	0.74	0.91
Poynten (2018) [33]	HSIL	0.04	0.99	0.71	0.54	0.04	0.99	0.71	0.54
Karita (2020) [31]	HSIL	0.20	0.99	0.75	0.85	0.04	0.994	0.20	0.93

Sens: Sensitivity. Sp: Specificity. PPV: positive predictive value. NPV: negative predictive positive.

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
