# Peer review of "Assessing the Potential of HPV16 E6 Seroprevalence as a Biomarker for Anal Dysplasia and Cancer Screening—A Systematic Review and Meta-Analysis"

_ijms, 2024, doi:10.3390/ijms25063437_

Round 1

Reviewer 1 Report

Comments and Suggestions for Authors

I was pleased to read this interesting systematic review and meta-analysis on the diagnostic performance of anal cancer. The manuscript is the newest in his field, is clearly structured and cites the most recent literature relevant to the topic. This is a current area of interest in neoplasia, thus, accurate screening algorithms for monitoring are crucial for an optimal management.

The conclusions are consistent with the purpose of this research, in order to provide an accurate and specific non-invasive biomarker tool for diagnostic purpose in a highly agressive skin tumor.  

Minor comments are as listed:

Ø  author affiliation are written incorrectly

Ø  all references are written without following the intructions

I recommend accepting after a minor revision.

Reviewer 2 Report

Comments and Suggestions for Authors

In this manuscript, Belmonte ST  and collaborators provided a systematic review and meta-analysis on HPV16 E6 as a potential biomarker for anal dysplasia and cancer screening. This analysis included six studies meeting inclusion criteria, assessing HPV16 E6 seroprevalence in individuals with anal HSIL or AC. Although HPV16 E6 seropositivity may not be an ideal standalone test for detecting anal cancer, HPV16 E6 seropositivity has been found to be a specific marker with potential clinical significance for a triage strategy.

Comments/suggestions:

Overall, this review is well organize and clearly written. As a review, could be nice schematically represent the HPV16 E6 structure and function in HPV-infected cells that is involved in anal cancer formation.

Reviewer 3 Report

Comments and Suggestions for Authors

This submission explores the potential use of HPV16 E6 seroprevalence (as assessed by using multiplex serology based on glutathione S-transferase fusion proteins in combination with fluorescent bead technology) in the triage of HPV16 driven anal cancer (AC) precursor lesions among at –risk individuals [with special focus on  HIV(+)/MSM’s]. Anal cancer represents a growing global health concern with severe morbidity and mortality among young individuals, especially PLHIV.

There is an Inconsistency in the affiliations; affiliation #3 is not linked with any of the distinguished author’s names.

In the Summary section, lines 28-30 commenting on the clinical utility of this biomarker are not totally in line with this assay’s proposed potential indications in the Conclusions section at the end of the manuscript.

Materials and Methods Section

Lines 102-103: “HPV16 E6 antibodies by using multiplex serology based on glutathione S-transferase fusion proteins in combination with fluorescent bead technology”. Many readers would be keen to know on the existence of commercially available or at least validated related assays.

In the US, the VLP anti-HPV vaccine has been FDA approved since 2010 for the prevention of anal Ca. The HPV vaccine is not mentioned anywhere in the text and the literature clearly suggests a clear benefit in incident anal HPV infections. An insight on how many studies included in this SLR & Meta-Analysis assessed HPV vaccination history as well as potential co-variations of HPV16 E6 seroprevalence with vaccination status would be most meaningful.

Statistics, tables and Figures are state of the art.

Results and Discussion

Both the “14.56 times higher probability of HPV16 E6 seropositivity in patients with HSIL+ compared to controls” (lines 170-171) and the “26.09-fold increase 177 (95%CI: 8.70, 78.28) in the probability of developing AC in individuals testing positive for 178 HPV16 E6, as compared to controls” (lines 170-171) are impressive and clinically relevant.

In lines 284-288 the authors correctly identify several factors which could explain much of the observed variations between studies, despite the high overall specificity of HPV16 E6 as a potential marker for HPV-related AC.

The Conclusion section is concise.

References are up-to date.
